# Triage-HF Validation in Heart Failure Clinical Practice: Importance of Episode Duration

**DOI:** 10.3390/diagnostics15121476

**Published:** 2025-06-10

**Authors:** Daniel García Iglesias, David Ledesma Oloriz, Diego Pérez Diez, David Calvo Cuervo, Rut Álvarez Velasco, Alejandro Junco-Vicente, José Manuel Rubín López

**Affiliations:** 1Arrhythmia Unit, Cardiology Department, Hospital Universitario Central de Asturias, 33011 Oviedo, Spain; 2Instituto de Investigación Sanitaria del Principado de Asturias, 33011 Oviedo, Spain; 3Cardiology Department, Hospital Universitario de Cabueñes, 33394 Gijón, Spain; 4Arrhythmia Unit, Cardiology Department, Hospital Clínico San Carlos, 28040 Madrid, Spain; 5Cardiology Department, Hospital Universitario de San Agustín, 33401 Avilés, Spain

**Keywords:** heart failure, implantable cardiac defibrillators, algorithm, TriageHF, Amplia DR, prediction

## Abstract

**Introduction:** The prevention of heart failure (HF) exacerbation is crucial for patient prognosis, and preventive treatment for potential symptoms and warning signs is essential in this context. The TriageHF © algorithm has been retrospectively validated and has demonstrated good correlation with HF episodes. This study analyzes the effectiveness of the TriageHF © algorithm in routine clinical practice, emphasizing the role of episode duration in its predictive capacity. **Materials and methods:** From October 2017 to October 2020, all patients who received a Medtronic Amplia DR implant were prospectively selected for analysis. To evaluate the algorithm’s diagnostic capacity, it was compared with the clinical diagnosis of HF episodes during follow-up. **Results:** The sustained moderate-risk (more than 7 days) and high-risk alerts both showed high positive predictive values (11.25% and 27.27%, respectively), along with an increase in the relative risk of HF, particularly in high-risk alerts (hazard ratio is 46.21 times higher than for sustained moderate-risk alerts). Furthermore, there was higher event-free survival in real low-risk alerts than in both sustained medium-risk and high-risk alerts (*p* < 0.01). **Conclusions:** TriageHF © can predict the worsening of patients with ICD CRT. Medium-risk alerts lasting less than 7 days do not pose a greater risk of HF episodes, while high-risk alerts, regardless of their duration, are highly correlated with HF episodes.

## 1. Introduction

Heart failure (HF) is a highly prevalent condition in the general population. Its diagnosis involves recognizing a set of signs and symptoms supported by some complementary tests [1]. Preventing exacerbations of HF is crucial for patient prognosis, and the early treatment of possible symptoms and warning signs is essential in this context [2].

Implantable cardiac defibrillators (ICDs) were initially developed as a treatment for malignant ventricular arrhythmias in patients with various heart conditions and a higher risk of sudden cardiac death [3,4,5,6]. They were later equipped with other functions, such as home monitoring functions [7,8,9,10,11,12]. Home monitoring has become a cornerstone in ICD follow-up since it has improved patient follow-up, allowing the early detection of complications and leading to better long-term outcomes. Given the positive outcomes of home monitoring in managing ICD-related complications during follow-up, its implementation is now globally recommended for all patients with ICDs [13].

In recent years, systems for the early diagnosis of HF based on ICD records have been developed. Initially, these systems for the early diagnosis of HF episodes were based on a single-sensed parameter [14,15,16,17]. However, they had a high rate of false-positive alerts, which significantly limited their clinical usefulness. As a result, algorithms have been developed to integrate multiple parameters related to HF, leading to a more accurate prediction of HF episodes [18,19,20]. Two such algorithms are currently available: the Boston HeartLogic © [21,22,23] and the Medtronic TriageHF © algorithms [24,25]. The TriageHF © algorithm has been retrospectively validated and has demonstrated good correlation with HF episodes in patients in other clinical studies [26,27]. In summary, the algorithm detects patients at higher risk of having an HF episode and rules it out in the absence of any risk alert based on its high negative predictive value. Moreover, it has been demonstrated that the use of these algorithms for heart failure monitoring, followed by timely interventions in patient management, translates into a significant clinical benefit [28,29,30,31].

Some recent systematic reviews clearly demonstrated that patients presenting high-risk readings not only had an increased risk of heart failure events but also a higher risk of all-cause mortality [32]. Moreover, it has been shown that these algorithms reduce the number of clinic visits and hospitalizations [33].

In previous studies, all alerts were considered from the beginning, regardless of their final duration (to date, the role of alert duration in the algorithm has not been evaluated). In this regard, the fact that an alert persists for more than a week may indicate an additional risk of HF that is not accounted for by the other parameters of the algorithm. Moreover, this algorithm has barely been evaluated in routine clinical practice, making external validation necessary to assess its predictive capability.

## 2. Objectives

The objectives of this study are to evaluate the usefulness of the TriageHF © algorithm for the early diagnosis of HF episodes, to validate its role in preventing HF episodes in routine clinical practice, and to assess the usefulness of episode duration in order to optimize the algorithm’s predictive capacity.

## 3. Materials and Methods

### 3.1. Study Population

*Inclusion criteria*: From October 2017 to October 2020, all patients referred to our arrhythmia unit with a diagnosis of heart failure, left ventricular dysfunction, and an indication for implantable cardioverter defibrillator–cardiac resynchronization therapy (ICD-CRT) who received a Medtronic (Minneapolis, MN, USA) Amplia DR implant (according to the clinical guidelines at the time) were prospectively selected for analysis.

*Exclusion criteria*: Patients who were unable to set up the remote monitoring transmitter at home were excluded from the study.

### 3.2. TriageHF © Algorithm Description

TriageHF © is an advanced algorithm designed to assist in the early identification of patients at risk of worsening HF. This predictive tool leverages data obtained from ICDs to provide a risk stratification framework that aids healthcare providers in timely clinical interventions. The TriageHF © algorithm continuously monitors multiple physiological and device-derived parameters that are indicative of worsening heart failure. These parameters include intrathoracic impedance (a marker of pulmonary congestion since decreasing impedance suggests fluid accumulation in the lungs), heart rate variability (a reduced HRV is often associated with increased autonomic dysfunction and worsening HF), patient activity levels (a decline in physical activity can be an early indicator of worsening functional status), night heart rates (elevated night-time heart rates may reflect increased sympathetic activity, which is linked to HF decompensation), and atrial and ventricular arrhythmias (the presence and frequency of arrhythmias can signal a worsening clinical condition). The algorithm processes these variables and assigns patients a risk classification based on their likelihood of experiencing HF deterioration. The risk levels are typically categorized into three tiers: low risk (patients exhibit stable physiological markers and do not show significant trends toward deterioration), medium risk (moderate changes in parameters that may warrant closer observation or minor adjustments in therapy), and high risk (significant deviations in monitored metrics, indicating a high likelihood of HF exacerbation, requiring immediate clinical attention).

### 3.3. Care Link Data Analytics and Alert Categorization

All patients were monitored using the Medtronic Home Monitoring (HM) system, which transmitted data every 3 months. The Carelink © platform provided HF monitoring data, which were blindly evaluated by two expert electrophysiologists to assess the device’s normal operation and HF risk events. Carelink © transmissions included TriageHF © alert information, which was further evaluated by two independent electrophysiologists.

TriageHF © alerts were later categorized as medium-risk or high-risk according to the system’s information. If a patient did not receive a medium-risk or high-risk alert in a given month, it was considered a low-risk alert. For each medium-risk or high-risk episode, its duration was evaluated specifically. A differentiation was made between medium-risk episodes, with episodes of less than one week categorized as medium-risk short-duration and episodes of one week or more as sustained medium-risk. Low-risk and medium-risk short-duration alerts were considered together as real low-risk alerts for further survival analyses.

### 3.4. Clinical Follow-Up

At the time of implantation, the clinical profile of each patient was recorded. Patients were followed up at one month after implantation and then every 6 months in the outpatient clinic. In addition, all patients were remotely monitored using the Medtronic Carelink © HM system, with transmissions every 3 months. At the end of the study, all patients were contacted by telephone to evaluate their final clinical status.

A composite endpoint of HF episodes was defined as a clinical event, which included the following clinical situations: HF mortality, hospitalization due to HF, consultation in the emergency room due to HF, or deterioration of the NYHA class that required any modification of treatment.

### 3.5. Statistical Analysis

Continuous variables were presented as mean and standard deviation, while categorical variables were presented as frequency and percentage. Student’s *t*-test was used for comparing continuous variables (after assessing the normality of continuous variables), and the Chi-square test was used for comparing categorical variables (a Yates modification was carried out, if needed) between patients with and without clinical events. All intervals were expressed with 95% confidence. A Kaplan–Meyer test was performed for survival analysis of the composite endpoint of HF episodes.

The diagnostic performance of the algorithm was evaluated by calculating the sensitivity, specificity, positive predictive value (PPV), and negative predictive value (NPV) of the algorithm in comparison with the clinical diagnosis of HF episodes at follow-up. Additionally, a survival analysis for the composite endpoint of HF episodes was conducted for each alert and month, comparing low-risk (including low-risk and medium-risk short-duration), medium-risk (sustained medium-risk), and high-risk alerts. For survival comparison between groups, a log-rank test was used, and to calculate the hazard ratio, a Cox regression was carried out. 

All statistical calculations were performed using R statistical software 3.4 [34].

### 3.6. Ethical Aspects

The study was approved by the ethics committee of our institution. All procedures were carried out in accordance with the standards and requirements of the Declaration of Helsinki and its subsequent amendments for human research. All data collected were handled in accordance with data protection regulations. Informed consent was obtained from all patients.

## 4. Results

### 4.1. Clinical Characteristics of the Study Population

During the study’s average follow-up period of 18 months, a total of 623 alert-months were analyzed. This corresponded to 211 medium-risk or high-risk alerts, originating from 39 different patients included in the study (Figure 1). The mean age of the patients was 67 years old, with males comprising 77% of the population. The main cause for ICD-CRT implantation was ischemic dilated cardiomyopathy (54%), mostly in primary prevention (85%). At the time of implantation, most patients were in NYHA class II (64%). More clinical details can be seen in Table 1.

### 4.2. Clinical Events in Follow-Up

During the follow-up period, 14 out of 39 patients (36%) experienced adverse clinical events, resulting in a total of 20 events. Of these events, two were HF-related deaths, seven were HF-related hospitalizations, four were emergency room visits due to HF, and seven were treatment modifications due to HF symptoms. The group with clinical events had a higher prevalence of atrial fibrillation (64% vs. 36%, *p* = 0.01), higher use of diuretic treatment (100% vs. 68%, *p* = 0.05), and lower use of sacubitril/valsartan treatment (0 vs. 36%, *p* = 0.03) than the group without clinical events (more details in Table 1).

### 4.3. Predictive Capacity of the TriageHF © Algorithm

During the follow-up period, 211 alerts were generated, with 189 (89.11%) being medium-risk alerts and 22 (10.89%) high-risk alerts. Among the medium-risk alerts, 35 were short-duration medium-risk alerts, lasting less than 7 days, while 154 were sustained medium-risk alerts, lasting a week or more. There were no medium-risk short-duration alerts resulting in HF clinical events.

Among the sustained moderate-risk alerts (lasting 7 days or more), there were 12 events, and among the high-risk alerts, there were 6 events. The diagnostic capacity of each group is shown in Table 2, and the sensitivity, specificity, PPV, and NPV values for the diagnosis of HF in both groups are shown in Table 3. The PPV was high for both sustained moderate-risk alerts (greater than 7 days) and high-risk alerts (11.25% and 27.27%, respectively), with an increase in the relative risk of HF when comparing both sustained medium-risk alerts with low-risk alerts (hazard ratio [HR] 24.23 [23.49–24.98]) and high-risk alerts with sustained moderate-risk alerts (HR 48.61 [47.79–49.43]).

The mean time to onset of the HF episodes was 13.67 days (+/− 4.72 days) for the high-risk alerts and 14.67 days (+/− 5.55 days) for the medium-risk alerts (*p* = 0.7). Survival analysis (Figure 2) showed significantly higher event-free survival throughout the month following the alert for real low-risk alerts than for both sustained medium-risk and high-risk alerts (*p* < 0.01) and also higher event-free survival for sustained medium-risk alerts than for high-risk alerts (*p* < 0.01).

Similar to the comparison between patients with and without HF decompensation events, patients with sustained moderate-risk and high-risk alerts showed a higher prevalence of atrial fibrillation (48.9% vs. 37.5%, *p* < 0.01). The remaining variables showed similar behavior between groups. A detailed comparison between these groups is presented in Table 4.

## 5. Discussion

Given the variety of available parameters and algorithms, as well as the range of outcomes obtained, a common question is when to consider an alert significant. In this regard, using standards with a high false-positive rate can lead to an excessive clinical workload, while employing algorithms with low sensitivity may result in underdiagnosis. For this reason, it is essential to thoroughly understand the performance of these algorithms in different scenarios, as well as the predictive capability of the various alert levels.

The TriageHF © algorithm is a multiparametric algorithm designed for the early diagnosis of HF episodes. This algorithm generates three types of alerts based on the risk of developing an HF episode within the next 30 days (low, intermediate, or high). In previous papers, it has shown good performance for HF prediction, but its routine use in clinical practice has not yet been extensively validated. In this regard, we designed the present study to assess the diagnostic capacity of the algorithm in routine clinical practice. Moreover, this work evaluated alert duration for the first time, which may significantly enhance the predictive capability of this algorithm. In addition, this improvement in the algorithm’s diagnostic capability may help reduce the rate of false positives in alerts without causing a loss in the algorithm’s sensitivity, which can have a significant impact on daily clinical practice.

Previous studies have described a high NPV for low-risk alerts (higher than 95%) [24,25,26,27]. In our study, during the 421 months in which patients were at low risk (low-risk alerts), there were only two events (0.48%) in the following 30 days after the alert. This high NPV (99.55%) confirms the low incidence of HF episodes in the following 30 days and suggests that these alerts can be ignored. On the other hand, our work is consistent with previous studies, which have considered high-risk alerts to be of real high risk, with a non-significant PPV [27] and a high HR for HF episodes [26]. Thus, these alerts need prompt attention and should not be ignored. For this reason, the overall results of our study, which are in line with previous publications, highlight two key messages: on the one hand, the high negative predictive value of low-risk alerts, and on the other, the high sensitivity of high-risk alerts.

Several studies have also been conducted based on the HeartLogic algorithm. Similar to those involving TriageHF, the initial investigations were primarily descriptive in nature. Although it is another alternative, these algorithms are not directly comparable due to the nature of each, and the utility of one or the other, in most cases, depends more on the availability of each device than on the peculiarities of each.

Integrating our findings with an AI-based system capable of filtering alerts and even diagnosing heart failure based on clinical scoring systems could significantly enhance the efficiency of algorithm-driven programs in terms of human and economic resource utilization.

The main finding of our study, not previously described in other clinical trials, is the importance of alert duration for medium-risk alerts. In previous works [24,25,26,27], medium-risk alerts were considered to be of low risk since a low number of clinical events were seen in the following 30 days after alert onset. However, our study highlights the importance of dividing medium-risk alerts into those lasting less or more than a week. Similar to low-risk alerts, we found that no clinical event was observed in the 30-day period following the alert onset for short-duration medium-risk alerts. Therefore, both types (low-risk and short-duration medium-risk alerts) could be grouped under the same category of real low-risk alerts and should not be taken into consideration. Moreover, this finding increases the specificity of the algorithm from 44% to 75.85%.

For sustained medium-risk alerts, we found a PPV of 11.25% and a relative risk of HF with respect to low-risk alerts of 24.23%. Thus, sustained medium-risk alerts should be treated similarly to high-risk alerts and handled with care, as a treatment modification may be necessary to avoid a possible HF event. In this sense, the previous literature has proposed several distinct protocols based on the TriageHF algorithm. Our findings support the incorporation of alert duration into home monitoring protocols, with the aim of improving the cost-effectiveness of these strategies by reducing the amount of false-positive records. Based on our findings, the logical next step would be to perform a comparative study between a pharmacological intervention protocol that excludes alerts shorter than seven days and one that includes them to assess the safety of this approach and its potential to reduce false-positive rates.

The cause of these findings may be multifactorial. On the one hand, the construction and design of the algorithm, based on multiple parameters, must be taken into account. In this regard, it is possible that marginal measurements of some of the variables may influence an increase in the risk level even if there is no actual increase in the patient’s clinical risk. In such situations—especially in moderate-risk cases—allowing time to determine whether it is a marginal measurement or a true alert may be useful. The fact that once these short-duration alerts are classified as low-risk, the diagnostic performance of the algorithm remains preserved would support this hypothesis.

On the other hand, the patient’s individual risk profile must be considered. Although the algorithm includes multiple variables, it is clear that some clinically relevant factors may not be captured by it. In this sense, the persistence of moderate-risk alerts over time could reflect a specific physiological background, indicating a higher baseline risk in the patient (likely due to some unmeasured variable). Therefore, the fact that a moderate-risk alert is not sustained may indicate a lower baseline risk, and such alerts could then be assimilated to a low-risk status.

In any case, the exact pathophysiological mechanism by which short-duration moderate-risk alerts behave similarly to low-risk alerts remains unknown. Thus, these hypotheses remain speculative. Further studies or additional measurements would be required to confirm the role of these hypotheses in the behavior of short-duration moderate-risk alerts.

The study analyzes a total of 623 alert-months, which come from 39 patients, and this small number of patients may be a potential limitation of this study. Although we have a limited patient sample, the method used to analyze the alerts results in a high number of alerts, which we believe contributes to the internal validity of the findings. It is true that these data are derived from a cohort of only 39 patients, which may affect the external validity of the results. Nevertheless, we believe our cohort is reasonably representative of the broader population of patients with heart failure.

None of the patients in the clinical events group was receiving sacubitril as part of their treatment, but 60% of the patients in the other group were. It is also worth noting the difference in the percentage of diuretic prescriptions between the two groups. It is possible that the first group of patients was in a more advanced stage of the disease and therefore unable to tolerate sacubitril treatment, which may partly explain the higher number of clinical events observed and the higher loop diuretic prescription. The other explication is that the effect of the sacubitril itself justifies the lower event rate in this group. However, this remains a hypothesis, as we do not have sufficient data to confirm it.

A significant difference in atrial fibrillation prevalence was observed (both when stratifying by the presence of device alerts and by the occurrence of clinical events). Regarding the incorporation of atrial fibrillation as an independent predictor of the reliability of a heart failure risk alert, our study does not provide sufficient data to confirm this hypothesis. Although it is well established that atrial fibrillation increases the risk of acute heart failure, the fact that both atrial fibrillation and a poorly controlled heart rate in the context of atrial fibrillation are among the multiple parameters assessed by the TriageHF^TM^ algorithm could contribute to an increased rate of false-positive alerts.

In the clinical events group, 9 out of 14 patients were classified as NYHA class III or higher, and 9 out of 14 had atrial fibrillation. Additionally, there were four deaths due to heart failure. Altogether, these findings suggest that this was a cohort of patients in an already advanced stage of the disease. The fact that these alerts occur in patients with a higher pretest probability of heart failure increases the likelihood of a subsequent clinical event, which could account for the higher hazard ratios observed.

## 6. Conclusions

Overall, the TriageHF © system shows promise in predicting the worsening of patients with ICD CRT and can potentially aid in their management. It is recommended that the system be routinely included in the care of these patients. However, it is important to note that medium-risk alerts lasting less than 7 days do not pose a significant risk of HF episodes and therefore should not generate unnecessary consultations. On the other hand, any high-risk alert, regardless of duration, is highly correlated with HF episodes and should be investigated promptly to prevent future HF events.

In conclusion, our findings could support the development of a new study that ignores short-duration alerts, thereby enabling a streamlined follow-up strategy involving a reduced number of alert reviews. This approach has the potential to optimize the use of human resources without compromising patient safety.

## Figures and Tables

**Figure 1 diagnostics-15-01476-f001:**
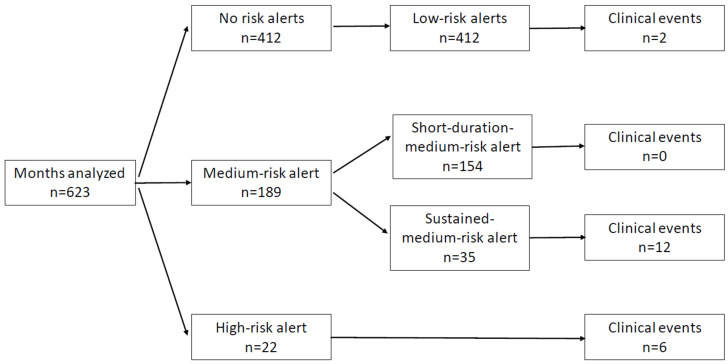
Alert flowchart with clinical events for each risk alert.

**Figure 2 diagnostics-15-01476-f002:**
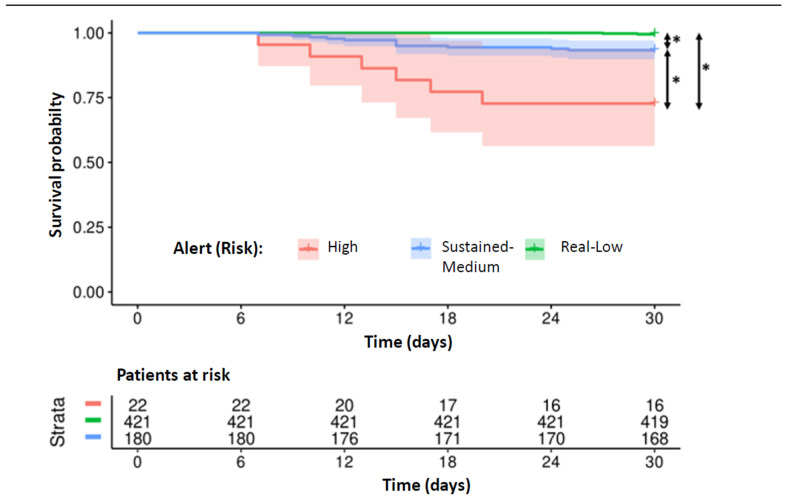
Analysis of 30-day survival comparing different Triage-HF alerts. Alerts are categorized as high risk, sustained medium risk, or low risk based on the criteria described in the text * *p* < 0.01.

**Table 1 diagnostics-15-01476-t001:** Clinical characteristics of the included patients. Basal and comparison between groups, with or without clinical events at follow-up.

Variable	Total Sample (N = 39)	Clinical Events (N = 14)	No Clinical Events (N = 25)	*p*
Clinical Features				
Sex	30 (76.92%)	11 (78.57%)	19 (76%)	1
Age (Years)	67.47 ± 9.91	71.01	65.495	0.1
Hypertension	22 (56.41%)	10 (71.43%)	12 (48%)	0.281
Diabetes mellitus	14 (35.9%)	5 (35.71%)	9 (36%)	1
Dyslipidemia	25 (64.1%)	8 (57.24%)	17 (68%)	0.741
Smoking	27 (56.41%)	8 (57.24%)	19 (76%)	0.389
Chronic pulmonary dis.	7 (17.95%)	3 (21.43%)	4 (16%)	1
Renal insufficiency	11 (28.21%)	5 (35.71%)	6 (24%)	0.683
Hepatopathy	2 (5.13%)	0 (0%)	2 (8%)	0.742
Sleep apnea	5 (12.82%)	3 (21.43%)	2 (8%)	0.481
Peripheral vasculopathies	8 (20.51%)	1 (7.14%)	7 (28%)	0.257
Ischemic DCM	21 (53.85%)	7 (50%)	14 (56%)	0.979
Primary prevention	33 (84.62%)	12 (85.71%)	21 (84%)	1
Atrial fibrillation	18 (46.15%) *	9 (64.29%)	9 (36%)	0.011
NYHA class II	22 (64.1%)	5 (35.71%)	17 (68%)	0.063
NYHA class III	14 (28.2%)	7 (50%)	7 (28%)	0.305
NYHA class IV	3 (7.7%)	2 (14.29%)	1 (4%)	0.596
Follow-up months	17.93 ± 9.81	21.56	15.89	0.116
Drug treatment				
Diuretics	31 (79.49%) *	14 (100%)	17 (68%)	0.05
ECAIs/ARBs	26 (66.67%)	11 (78.57%)	15 (60%)	0.409
Sacubutril	9 (23.08%) *	0 (0%)	9 (36%)	0.03
Beta-blockers	36 (92.31%)	12 (85.71%)	24 (96%)	0.596
AMRs	11 (28.21%)	2 (14.29%)	9 (36%)	0.283
Clinical events				
Death	2 (5.13%) *	4 (28.57%)	0 (0%)	0.023
Hospital admission	7 (17.95%) *	7 (50%)	0 (0%)	0.001
Emergency consultation	4 (10.26%) *	4 (28.57%)	0 (0%)	0.023
Outpatient consultation	7 (17.95%) *	4 (28.57%)	0 (0%)	0.023

DCM: Dilated cardiomyopathy; NYHA: New York Heart Association; ECAIs: Angiotensin-converting enzyme inhibitors; ARBs: Angiotensin II receptor blockers; AMRs: Mineralocorticoid receptor antagonist. * Differences are statistically significant.

**Table 2 diagnostics-15-01476-t002:** Evaluation of the algorithm’s diagnostic capacity comparing diagnosed HF events at follow-up.

	HF Clinical Event	No HF Clinical Event	HR	*p*
Low-risk	2 (0.48%)	419 (99.52%)		
Medium-risk short-duration	0 (0%)	35 (100%)	18.45 (17.71–19.2)	
Sustained medium-risk	12 (7.79%)	142 (92.21%)	24.23 (23.49–24.98)	<0.001
High-risk	6 (27.27%)	16 (72.73%)	72.84 (72.02–73.66)	<0.001

**Table 3 diagnostics-15-01476-t003:** Evaluation of sensitivity, specificity, positive predictive value, and negative predictive value of the different types of alerts.

	Sustained Medium-Risk	High-Risk
Sensitivity (%)	90	30
Specificity (%)	75.85	97.35
PPV (%)	11.25	27.27
NPV (%)	99.55	97.67

PPV: positive predictive value; NPV: negative predictive value.

**Table 4 diagnostics-15-01476-t004:** Clinical characteristics of the included patients. Comparison between groups with or without TriageHF risk alerts.

	TriageHF Alerts		No TriageHF Alerts		*p*
	N = 31		N = 8		
	Mean/Num.	Std. Dev./Perc.	Mean/Num.	Std. Dev./Perc.	
Clinical Features					
Sex (male)	25	80.65	5	62.5	0.538
Age (years)	66.65	10.116	70.65	8.93	0.29
High blood pressure	18	58.06	4	50	0.992
Diabetes mellitus	13	41.94	1	12.5	0.257
Dyslipidemia	19	61.29	6	75	0.759
Smoking	21	67.74	6	75	1
Chronic pulmonary dis.	4	12.9	3	37.5	0.272
Renal insufficiency	9	29.03	2	25	1
Hepatopathy	2	6.45	0	0	1
Sleep apnea	3	9.68	2	25	0.574
Peripheral vasculopathies	7	22.58	1	12.5	0.89
Ischemic DCM	18	58.06	3	37.5	0.521
Primary prevention	27	87.1	6	75	0.767
Atrial fibrillation	15	48.39	3	37.5	0.001
NYHA class III	9	29.03	5	62.5	0.178
NYHA class IV	2	6.45	1	12.5	1
Follow-up (months)	19.18	9.07	13.05	11.67	0.199
Drug Treatment					
Diuretics	23	74.19	8	100	0.262
ECAIs/ARBs	22	70.97	4	50	0.483
Sacubitril	7	22.58	2	25	1
Beta-blockers	29	93.55	7	87.5	1
AMRs	9	29.03	2	25	1

Comparison between patients with TriageHF alerts (high-risk or sustained medium-risk) and patients without TriageHF alerts (low-risk or short-duration medium-risk alerts). DCM: Dilatated cardiomyopathy; NYHA: New York Heart Association; ECAIs: Angiotensin-converting enzyme inhibitors; ARBs: Angiotensin II receptor blockers; AMRs: Mineralocorticoid receptor antagonist.

## Data Availability

The data presented in this study are available on request from the corresponding author.

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
