# Peer review of "Triage-HF Validation in Heart Failure Clinical Practice: Importance of Episode Duration"

_diagnostics, 2025, doi:10.3390/diagnostics15121476_

Round 1

Reviewer 1 Report (Previous Reviewer 2)

Comments and Suggestions for Authors

Hello,
As the revised manuscript has addressed all the major concerns and reviewer has reverted with accept decision, we may go ahead with acceptance after editorial desk acceptance.
Thanks

Author Response

Thanks for all your valuable comments in previous revisions

Reviewer 2 Report (New Reviewer)

Comments and Suggestions for Authors

Thank you for an interesting research article.

  1. Why you did not mention how did you calculate the hazards ratio in your statistical analysis? Was it based on the COX regression or some other simplified formula?
  2. Table 1 is very confusing. Better to report as Mean ± Standard Deviation and Number (Percentage) and why not provide a matching univariate analysis between the two groups: low risk and short medium risk versus sustained medium risk and high risk?
  3. In Table 2, NYHA Class II is 2.643 (Is this a mean or a number?) How come? I think it should be a number (Count)
  4. Please be consistent in using decimals as one time you use 0.000 sometimes 0.00 or 0.0 or no decimals at all, please unify the number of decimals. Example Tables 1 and 2.
  5. This statement “There were no medium-risk-short-duration alerts 176 resulting in any HF clinical event during low-risk alerts.” In line 176-177 the phrase “during low-risk alerts” is redundant.
  6. Why do you keep repeating sustained medium risk (lasting more than 7 days) no need to keep repeating the phrase in the brackets, you already defined it?
  7. Table 2 and 5 are very confusing. I understood that Table 2 comparing patients who had clinical events with those who did not have clinical events. Table 5 is comparing patients who had alerts to those who had the alerts. But again, you used in the top row clinical events versus no clinical events!!
  8. The first two paragraphs in discussion are repeated from introduction (background) section.
  9. There are major problems in this article which can be summarized as follows: First, the concept in the study is a simple concept of duration of alerts or risks, so if they are medium lasting more than 7 days, these are important alerts and are associated with more HF exacerbations and adverse outcomes. The paper keeps repeating this simple idea and even it is mentioned in the introduction ahead of the research in line 60 – 61. A better approach would be to make the article more like a story and wait until the results are presented to make that statement. The second point is that the statistics are not well presented. Again, explain how you performed the HR calculation, and instead of having 3 tables, one for the entire patients, and then for clinical versus no clinical event and finally for those with alerts and those without alerts, sum it up in one or two tables only. For example, you could have total versus, no or low risk alert, and with sustained medium or high alert group. The second table can be a comparison of those with clinical event and those without a clinical event. Once you do these baselines you can then delve more into the parameters and which ones predict a sustained medium risk for example which you did not do here and this is a limitation. Readers want to know how did you know that duration of the medium risk was identified and why 7 why not less or more? How did you reach to this number? The third thing I struggled with here was the English language and repetitions and redundancies and I have given few comments on that above. And why for example you did not show a comparison in all variables between short medium and sustained medium risks?
  10. Also there has been previous external validation of TriageHF in several papers including as an example: Cardoso I, Cunha PS, Laranjo S, et al. Validation of a Heart Failure Risk Score in a Cohort of Cardiac Resynchronization Therapy Patients Under Remote Monitoring: Results from the TriageHF™ Algorithm. J Innov Card Rhythm Manag. 2023;14(9):5576-5581. Published 2023 Sep 15. doi:10.19102/icrm.2023.14093 So the external validation part is not novel
  11. Again, there is another group who have used a Triage HF plus algorithm where medium alerts were assessed and by that they performed well in detecting the cases that need intervention. Here the reference: Ahmed FZ, Taylor JK, Green C, et al. Triage-HF Plus: a novel device-based remote monitoring pathway to identify worsening heart failure. ESC Heart Fail. 2020;7(1):107-116. doi:10.1002/ehf2.12529 And although you mentioned it as a Triage HF not Triage HF plus you did not elaborate on its use to stratify the medium risk cases
  12. Also, you did not discuss the major differences between the groups in terms of A. Fib, diuretics and Sacubitril use. This is very important to discuss I believe. Like discuss how A. Fib is linked here, could it be that instead of the duration use the A. Fib as a marker of need to attend to the medium risk alerts?

Author Response

Thank you for an interesting research article. 

  1. Why you did not mention how did you calculate the hazards ratio in your statistical analysis? Was it based on the COX regression or some other simplified formula?  

To calculate the hazard ratio, a Cox regression analysis was conducted. 

  1. Table 1 is very confusing. Better to report as Mean ± Standard Deviation and Number (Percentage) and why not provide a matching univariate analysis between the two groups: low risk and short medium risk versus sustained medium risk and high risk?  

As recomended later it has been corrected (merging table 1 and 2).  

  1. In Table 2, NYHA Class II is 2.643 (Is this a mean or a number?) How come? I think it should be a number (Count)  

It was a typographic error and was corrected in the table.  

  1. Please be consistent in using decimals as one time you use 0.000 sometimes 0.00 or 0.0 or no decimals at all, please unify the number of decimals. Example Tables 1 and 2.  

<0.001 term was used.  

  1. This statement “There were no medium-risk-short-duration alerts 176 resulting in any HF clinical event during low-risk alerts.” In line 176-177 the phrase “during low-risk alerts” is redundant.  

It was corrected. 

  1. Why do you keep repeating sustained medium risk (lasting more than 7 days) no need to keep repeating the phrase in the brackets, you already defined it?  

We keep repeating it to clearly differentiate them from the short duration medium risk alerts because a previous reviewer recommended this. 

  1. Table 2 and 5 are very confusing. I understood that Table 2 comparing patients who had clinical events with those who did not have clinical events. Table 5 is comparing patients who had alerts to those who had the alerts. But again, you used in the top row clinical events versus no clinical events!!  

 It was corrected in Table 5. Furthermore, Table 2 was merged with Table 1, as suggested in previous comments. 

  1. The first two paragraphs in discussion are repeated from introduction (background) section.  

They were deleted from discussion to avoid redundancies.  

  1. There are major problems in this article which can be summarized as follows: First, the concept in the study is a simple concept of duration of alerts or risks, so if they are medium lasting more than 7 days, these are important alerts and are associated with more HF exacerbations and adverse outcomes. The paper keeps repeating this simple idea and even it is mentioned in the introduction ahead of the research in line 60 – 61. A better approach would be to make the article more like a story and wait until the results are presented to make that statement. The second point is that the statistics are not well presented. Again, explain how you performed the HR calculation, and instead of having 3 tables, one for the entire patients, and then for clinical versus no clinical event and finally for those with alerts and those without alerts, sum it up in one or two tables only. For example, you could have total versus, no or low risk alert, and with sustained medium or high alert group. The second table can be a comparison of those with clinical event and those without a clinical event. Once you do these baselines you can then delve more into the parameters and which ones predict a sustained medium risk for example which you did not do here and this is a limitation. Readers want to know how did you know that duration of the medium risk was identified and why 7 why not less or more? How did you reach to this number? The third thing I struggled with here was the English language and repetitions and redundancies and I have given few comments on that above. And why for example you did not show a comparison in all variables between short medium and sustained medium risks?  

HR calculation was introduced in methodology section as recommended for a previous reviewer.  

English language was corrected as suggested.  

Duration limit was established based on RE-heart study (included in introduction and references).  

As suggested previously, tables were reconstructed.  

  1. Also there has been previous external validation of TriageHF in several papers including as an example: Cardoso I, Cunha PS, Laranjo S, et al. Validation of a Heart Failure Risk Score in a Cohort of Cardiac Resynchronization Therapy Patients Under Remote Monitoring: Results from the TriageHF™ Algorithm. J Innov Card Rhythm Manag. 2023;14(9):5576-5581. Published 2023 Sep 15. doi:10.19102/icrm.2023.14093 So the external validation part is not novel  

The mentioned paper validates the capability for unplanned hospital admission due to HF decompensation prediction. But it only assesses the difference in hazard ratios between medium-risk and high-risk alerts was confirmed, not considering episode duration, which we suggest is a strong risk marker. 

  1. Again, there is another group who have used a Triage HF plus algorithm where medium alerts were assessed and by that they performed well in detecting the cases that need intervention. Here the reference: Ahmed FZ, Taylor JK, Green C, et al. Triage-HF Plus: a novel device-based remote monitoring pathway to identify worsening heart failure. ESC Heart Fail. 2020;7(1):107-116. doi:10.1002/ehf2.12529 And although you mentioned it as a Triage HF not Triage HF plus you did not elaborate on its use to stratify the medium risk cases  

In the TriageHF Plus the authors combined the algorithm alarm with a phone call. As recommended for a previous reviewer, it has been included in introduction section.  

  1. Also, you did not discuss the major differences between the groups in terms of A. Fib, diuretics and Sacubitril use. This is very important to discuss I believe. Like discuss how A. Fib is linked here, could it be that instead of the duration use the A. Fib as a marker of need to attend to the medium risk alerts?  

None of the patients in the clinical events group was receiving sacubitril as part of their treatment, meanwhile 60% of the patients in the other group was. It is also worth noting the difference in the percentage of diuretic prescriptions between the two groups. It is possible that the first group of patients was in a more advanced stage of the disease and therefore unable to tolerate sacubitril treatment, which may partly explain the higher number of clinical events observed and the higher loop diuretic prescription. The other explication is that the effect of the sacubitril itself justifies the lower event rate in this group. However, this remains a hypothesis, as we do not have sufficient data to confirm it.  

A significant difference in Atrial fibrillation prevalence was observed (both when stratifying by the presence of device alerts and by the occurrence of clinical events). Regarding the incorporation of atrial fibrillation as an independent predictor of the reliability of a heart failure risk alarm, our study does not provide sufficient data to confirm this hypothesis. Although it is well established that atrial fibrillation increases the risk of acute heart failure, the fact that both atrial fibrillation and poorly controlled heart rate in the context of atrial fibrillation are among the multiple parameters assessed by the TriageHF™ algorithm could contribute to an increased rate of false-positive alerts.    

Reviewer 3 Report (New Reviewer)

Comments and Suggestions for Authors

The article presents an important study investigating the usability of the TriageHF algorithm in heart failure (HF) clinical practice and, in particular, the predictive value of the duration of alerts. The method used is prospective patient follow-up and retrospective algorithm analysis. In the study, 39 patients with Medtronic Amplia DR implants were followed between 2017 and 2020 and a total of 623 “alert-months” were evaluated. Statistical analyses (Student's t-test, Chi-square test, Kaplan-Meier, log-rank test) were used in the study. The algorithm used was compared with a multi-parameter structure (intrathoracic impedance, heart rate variability, activity level, etc.) and clinical events, and a new classification based on the duration of medium-risk alerts (medium-risk alerts with a duration of less than 7 days were included in the low-risk group) was made. 39 patients, 211 medium-high risk alerts and 20 clinical events were reported. Although limited in terms of sample size, this is in line with similar studies in the literature.

Recommendations:

  • In the method section, the selection criteria and exclusion criteria of patients according to the clinical protocol should be clarified.
  • The effect of differences in patients' drug treatment (especially Sacubitril/Valsartan usage rates) on the algorithm performance should be discussed.
  • A strong association between high-risk warnings and HF events (HR 48.61).
  • References to some recent meta-analyses and systematic reviews can be added to the literature review.
  • Since the sample size is limited, the generalizability of the findings may be limited. This should be emphasized more clearly in the discussion section.
  • Comparisons with the literature should be made more systematically; similarities and differences of the results can be compared in more detail, especially with alternative algorithms such as HeartLogic.
  • A small paragraph can be devoted to comparisons with artificial intelligence and machine learning-based systems in the discussion section.
  • Recommendations for integration into clinical practice can be emphasized more strongly in the results.
  • Future development directions of the algorithm (e.g., artificial intelligence integration, personalized thresholds) can be discussed.
  • Limitations of the sample size should be clearly stated.
  • Literature comparisons should be made more extensively and new sources should be added.
  • Clinical integration suggestions and future research directions should be strengthened in the discussion section.

Author Response

The article presents an important study investigating the usability of the TriageHF algorithm in heart failure (HF) clinical practice and, in particular, the predictive value of the duration of alerts. The method used is prospective patient follow-up and retrospective algorithm analysis. In the study, 39 patients with Medtronic Amplia DR implants were followed between 2017 and 2020 and a total of 623 “alert-months” were evaluated. Statistical analyses (Student's t-test, Chi-square test, Kaplan-Meier, log-rank test) were used in the study. The algorithm used was compared with a multi-parameter structure (intrathoracic impedance, heart rate variability, activity level, etc.) and clinical events, and a new classification based on the duration of medium-risk alerts (medium-risk alerts with a duration of less than 7 days were included in the low-risk group) was made. 39 patients, 211 medium-high risk alerts and 20 clinical events were reported. Although limited in terms of sample size, this is in line with similar studies in the literature. 

Recommendations: 

  • In the method section, the selection criteria and exclusion criteria of patients according to the clinical protocol should be clarified.  

Inclusion criteria: From October 2017 to October 2020, all patients referred to our arrhythmia unit with a diagnosis of heart failure, left ventricular dysfunction, and an indication for implantable cardioverter defibrillator-cardiac resynchronization therapy (ICD-CRT,  who received a Medtronic Amplia DR implant (according to the clinical guidelines at the time), were prospectively selected for analysis.  

Exclusion criteria: Patients who were unable to set up the remote monitoring transmitter at home were excluded from the study. 

  • The effect of differences in patients' drug treatment (especially Sacubitril/Valsartan usage rates) on the algorithm performance should be discussed 

None of the patients in the clinical events group was receiving sacubitril as part of their treatment, meanwhile 60% of the patients in the other group was. It is also worth noting the difference in the percentage of diuretic prescriptions between the two groups. It is possible that the first group of patients was in a more advanced stage of the disease and therefore unable to tolerate sacubitril treatment, which may partly explain the higher number of clinical events observed and the higher loop diuretic prescription. The other explication is that the effect of the sacubitril itself justifies the lower event rate in this group. However, this remains a hypothesis, as we do not have sufficient data to confirm it. 

  • A strong association between high-risk warnings and HF events (HR 48.61).  

In the clinical events group, 9 out of 14 patients were classified as NYHA class III or higher, and 9 out of 14 had atrial fibrillation. Additionally, there were four deaths due to heart failure. Altogether, these findings suggest that this was a cohort of patients in an already advanced stage of the disease. The fact that these alerts occur in patients with a higher pretest probability of heart failure increases the likelihood that they will be followed by a clinical event, which could account for the higher hazard ratios observed. 

  • References to some recent meta-analyses and systematic reviews can be added to the literature review.  

Some recent systematic reviews clearly demonstrated that patients presenting high-risk readings not only had an increased risk of heart failure events, but also a higher risk of all-cause mortality [A]. Moreover, it is also showed that they reduce the number of clinic visits and hospitalizations 

[A] Ahmed FZ, Sammut-powell C, Kwok CS, Tay T, Motwani M, Martin GP, et al. Remote CIED monitoring Remote monitoring data from cardiac implantable electronic devices predicts all-cause mortality. Europace. 2022;245–55.  

[B] Andrea Tedeschi, Matteo Palazzini, Giancarlo Trimarchi, Nicolina Conti, Francesco Di Spigno et al. Failure Management through Telehealth: Expanding Care and Connecting Hearts.  J Clin Med. 2024 Apr 28;13(9):2592.  

  • Since the sample size is limited, the generalizability of the findings may be limited. This should be emphasized more clearly in the discussion section.  

Although we have a limited patient sample, the method used to analyze the alerts results in a high number of alarms, which we believe contributes to the internal validity of the findings. It is true that the fact these data are derived from a cohort of only 39 patients may affect the external validity of the results. Nevertheless, we believe our cohort is reasonably representative of the broader population of patients with heart failure. 

  • Comparisons with the literature should be made more systematically; similarities and differences of the results can be compared in more detail, especially with alternative algorithms such as HeartLogic.  

Several studies have also been conducted based on the HeartLogic algorithm. Similar to those involving TriageHF, the initial investigations were primarily descriptive in nature.  Although it is another alternative, both algorithms are not directly comparable due to the nature of each and the utility of one or the other in most cases depends more on the availability of each device rather than the peculiarities of each. 

  • A small paragraph can be devoted to comparisons with artificial intelligence and machine learning-based systems in the discussion section.  

Integrating our findings with an AI-based system capable of filtering alerts and even diagnosing heart failure based on clinical scoring systems could significantly enhance the efficiency of algorithm-driven programs in terms of human and economic resource utilization. 

  • Recommendations for integration into clinical practice can be emphasized more strongly in the results.  

Previous literature has proposed several distinct protocols based on the TriageHF algorithm. Our findings support the incorporation of alarm duration into home monitoring protocols, with the aim of improving the cost-effectiveness of these strategies by reducing the amount of false positive records. 

  • Future development directions of the algorithm (e.g., artificial intelligence integration, personalized thresholds) can be discussed.  

Based on our findings, the logical next step would be to perform a comparative study between a pharmacological intervention protocol that excludes alerts shorter than seven days and one that includes them, to assess the safety of this approach and its potential to reduce false positive rates.  

  • Limitations of the sample size should be clearly stated. 

It has been clarified previously.  

  • Literature comparisons should be made more extensively and new sources should be added.  

It was responded in a previous issue.  

  • Clinical integration suggestions and future research directions should be strengthened in the discussion section.  

In conclusion, our findings could support the development of a new study that ignores short-duration alerts, thereby enabling a streamlined follow-up strategy involving a reduced number of alert reviews. This approach has the potential to optimize the use of human resources without compromising patient safety. 

Round 2

Reviewer 2 Report (New Reviewer)

Comments and Suggestions for Authors

None

This manuscript is a resubmission of an earlier submission. The following is a list of the peer review reports and author responses from that submission.

Round 1

Reviewer 1 Report

Comments and Suggestions for Authors

This article provides a detailed analysis of the validation of Triage-HF in clinical practice for heart failure, emphasizing the importance of episode duration. The study results offer valuable guidance for clinical decision-making.

I have some comments:

 1. The discussion on the research motivation and contributions could be clearer, as it currently lacks depth and specificity.

 2. Issues with the formatting of references. References 46, 50, and 52 contain extra spaces in their formatting, for example: [14,15,16, 17] (there is a space before "17").

 3. Clarification on sample size and alert months. Could you explain why the study included only 39 patients but analyzed 623 alert months? Would it be possible to increase the sample size to improve statistical robustness?

 4. The study relies on multiple parameters (e.g., heart rate variability and activity levels), but it does not analyze the contribution of individual parameters. Would it be beneficial to assess which parameters have the greatest impact on heart failure prediction?

Author Response

This article provides a detailed analysis of the validation of Triage-HF in clinical practice for heart failure, emphasizing the importance of episode duration. The study results offer valuable guidance for clinical decision-making. 

I have some comments: 

  1. The discussion on the research motivation and contributions could be clearer, as it currently lacks depth and specificity.

Throughout the discussion, there are references to the justification of the study, based on the importance of reducing the rate of false positive results. For example:  

“Given the variety of available parameters and algorithms, as well as the range of outcomes obtained, a common question is when to consider an alert significant or not. In this regard, using standards with a high false positive rate can lead to an excessive clinical workload, while employing algorithms with low sensitivity may result in underdiagnosis. For this reason, it is essential to thoroughly understand the performance of these algorithms in different scenarios, as well as the predictive capability of the various alert levels.” 

In any case, a paragraph has been added as follows to clarify this: 

“Moreover, this improvement in the algorithm's diagnostic capability may help reduce the rate of false positives in alerts, without causing a loss in the algorithm's sensitivity, which can have a significant impact on daily clinical practice.” 

  1. Issues with the formatting of references. References 46, 50, and 52 contain extra spaces in their formatting, for example: [14,15,16, 17] (there is a space before "17").

It has been corrected  

  1. Clarification on sample size and alert months. Could you explain why the study included only 39 patients but analyzed 623 alert months? Would it be possible to increase the sample size to improve statistical robustness?

The study included a total of 39 patients, and each patient generated an average of 16 alerts per month, resulting in a rate of 623 alerts per month. Although the number of patients is limited, this has been offset by longer-term follow-up, which translates into a higher number of alerts per patient per month. The limited number of patients has been included as a limitation in the discussion: 

Although the study analyzes a total of 623 alerts-month, these come from 39 patients, and this small number of patients may be a potential limitation of the study. 

  1. The study relies on multiple parameters (e.g., heart rate variability and activity levels), but it does not analyze the contribution of individual parameters. Would it be beneficial to assess which parameters have the greatest impact on heart failure prediction?

The reviewer rightly points out that it could be useful to analyze which parameter triggers each alert; however, due to the proprietary nature of the algorithm, this could not be analyzed.  

Reviewer 2 Report

Comments and Suggestions for Authors

Hello,

Thanks for submitting this manuscript to this journal and giving me opportunity to share my review for title "Triage-Hf Validation in Heart Failure Clinical Practice. Importance of Episode Duration".

I have following suggestions-

  1. Since this algorithm is specific to one of the companies making ICD-CRT, authors should disclose any financial or other obligations for the study.
  2. What are the cut-offs adopted for low risk, moderate risk and high risk for the total score or individual variables analysed for the algorithm
  3. Please share statistical analysis and comparison of baseline characteristics between the 3 score groups as these may confound the outcomes 
  4. 4. Please share statistical analysis comparing low, moderate short, moderate sustained and high risk for confounders and whether event rates were different and statistically significant after propensity matching for confounders.
  5. Thanks 

Author Response

Thanks for submitting this manuscript to this journal and giving me opportunity to share my review for title "Triage-Hf Validation in Heart Failure Clinical Practice. Importance of Episode Duration". 

I have following suggestions: 

  1. Since this algorithm is specific to one of the companies making ICD-CRT, authors should disclose any financial or other obligations for the study.

There was no funding from any company, and the authors declare in the text that they have no conflicts of interest related to the present study. 

  1. What are the cut-offs adopted for low risk, moderate risk and high risk for the total score or individual variables analysed for the algorithm

The reviewer rightly points out that it could be useful to analyze which parameter triggers each alert; however, due to the proprietary nature of the algorithm, this could not be analyzed. Furthermore, the risk levels (low, medium, or high) are predefined and cannot be modified. 

  1. Please share statistical analysis and comparison of baseline characteristics between the 3 score groups as these may confound the outcomes.

Addressed along with point 4. 

  1. Please share statistical analysis comparing low, moderate short, moderate sustained and high risk for confounders and whether event rates were different and statistically significant after propensity matching for confounders.

Since patients may present alerts of all three types, it is not possible to analyze each variable separately based on a specific group of alerts. However, it is possible to analyze them based on whether or not they have presented any risk alerts. A table (table 5) has been added to complete the statistical analysis, as requested. 

Round 2

Reviewer 1 Report

Comments and Suggestions for Authors

The study includes only 39 patients, which limits statistical power and external validity.

The discussion section merely reiterates the results without providing an explanation.

Reviewer 2 Report

Comments and Suggestions for Authors

Hello,

thanks for replying to the queries and clarifying most of the doubts.

However, the major concern remains about finding difference between low, medium and high score group. Authors have reverted by commenting that a single patient may have more than 1 type of event. However, each patient rather than event can be catergorized as low or moderate or high risk rather than scoring individual events and then the utility of the model can be evaluated against it.

Apart from this rest of the queries have been addressed. Thanks
